# Nursing Students’ Perceptions and Experiences of Aggression During Clinical Placements

**DOI:** 10.3390/nursrep15070245

**Published:** 2025-07-02

**Authors:** Chaxiraxi Bacallado-Rodríguez, Francisco Javier Castro-Molina, Jesús Manuel García-Acosta, Silvia Elisa Razetto-Ramos, Vicente Llinares-Arvelo, José Ángel Rodríguez-Gómez

**Affiliations:** 1Canary Islands Health Service, 38071 Tenerife, Spain; cbacalla@ull.edu.es (C.B.-R.); fcastrom@ull.edu.es (F.J.C.-M.); srazetto@ull.edu.es (S.E.R.-R.); vllinare@ull.edu.es (V.L.-A.); 2Nuestra Señora de Candelaria School of Nursing, University of La Laguna, 38010 Santa Cruz de Tenerife, Spain; 3Faculty of Nursing, University of La Laguna, 38200 San Cristóbal de La Laguna, Spain; jarogo@ull.edu.es

**Keywords:** workplace violence, aggression, students, nursing, health personnel, clinical placement

## Abstract

**Background:** Violence against healthcare professionals is a growing public health concern. In Spain, the National Observatory of Aggressions recorded 16,866 cases in 2024, marking a 103.06% increase since 2017. This phenomenon has intensified in recent years, with serious repercussions for both the physical and psychological well-being of healthcare workers, as well as for the quality of care provided. **Objectives:** This descriptive study examines the knowledge, perceptions, and experiences of workplace aggression among undergraduate students at the University School of Nursing of the Nuestra Señora de Candelaria University Hospital. **Materials and Methods:** A self-administered ad hoc questionnaire was distributed to 266 students across all academic years to assess their knowledge and to explore their perceptions and experiences of aggression witnessed or experienced during clinical placements. This study was guided by the Strengthening the Reporting of Observational Studies in Epidemiology (STROBE) guidelines. **Results:** The findings revealed significant educational gaps among students regarding how to manage aggressive situations, as well as high levels of concern and an aggression exposure rate exceeding 30%. A statistically significant association was also observed in relation to the academic year. **Conclusions:** This study provides a foundation for the development of specific training programmes tailored to the needs identified and for enhancing occupational safety in healthcare settings.

## 1. Introduction

Aggression toward healthcare professionals constitutes a growing and concerning public health issue worldwide [1,2]. The World Health Organisation (WHO), together with the International Labour Organisation (ILO) and the International Council of Nurses (ICN), defines workplace violence as ‘incidents where staff are abused, threatened or assaulted in circumstances related to their work, including commuting to and from work, involving an explicit or implicit challenge to their safety, well-being or health’ [1].

The WHO reports that 62% of healthcare workers have experienced some form of workplace violence [3]. A meta-analysis conducted in 2020, which included data from 30 countries (18 in Europe, 17 in the Eastern Mediterranean region, 14 in the Western Pacific, 10 in the Americas, 4 in Africa, and 2 in South-East Asia), and comprising a total sample of 61,800 healthcare professionals, revealed that Europe had the highest prevalence of physical workplace violence, at 26.38%, followed by the Americas with 23.61%. This violence was perpetrated by patients themselves, as well as by their relatives and/or companions. Of the 65 studies included in the meta-analysis, 34 focused specifically on the nursing profession, highlighting the particular vulnerability of this group to such aggression [4]. Data from the ICN indicate that 86.2% of national nursing associations have reported incidents of violence perpetrated by patients or their relatives and/or companions [5]. In Spain, the National Observatory of Aggressions (ONA) recorded 16,866 reported cases in 2024, reflecting a 103% increase since 2017. Nurses represent the second most affected professional group, accounting for 28.97% of all reported aggressions [6]. This phenomenon has intensified in recent years, with significant repercussions for both the physical and psychological health of healthcare workers, as well as for the quality of care provided [7,8].

In Spain, the majority of reported aggressions in 2024 were non-physical (84%) as opposed to physical (16%) [5], a pattern also observed in international studies [9,10]. According to the findings of Fernández-Fernández et al., 26.5% of surveyed nursing students reported having been subjected to intimidation and/or harassment during their clinical placements in the past year, with foreign-born students experiencing these incidents more frequently [11]. The geographical distribution of reported cases within Spain identified Andalusia, the Valencian Community, and the Canary Islands as the regions with the highest incidence [6].

Since the reform of the Spanish Criminal Code in 2015, aggression against healthcare professionals has been classed as an offence of assault against authority [12], reinforcing both the legal recognition of the seriousness of such acts and the need for preventive measures. Available evidence highlights the importance of education in the prevention and management of potentially violent situations as a means to reduce the incidence of aggression [1,13,14,15,16,17,18,19].

Despite the relevance of the phenomenon, the Libro Blanco (White Paper) of the nursing profession [20] does not include specific content on workplace aggression or coping strategies, thus revealing a gap between the realities of clinical practice and the academic training provided to students.

Against this background, it is essential to examine nursing students’ perceptions and experiences of aggression, as they are future professionals who will inevitably face this reality upon entering the workforce.

## 2. Materials and Methods

### 2.1. Aims

The general aim of this study was to assess perceptions of aggression towards healthcare professionals and to analyse the experiences of aggression, either witnessed or experienced, during clinical placements.

The study also pursued the following specific objectives:a.To explore general knowledge regarding workplace aggression.b.To identify potential associations between perceptions and experiences of aggression and sociodemographic variables.

### 2.2. Study Design and Setting

This research adhered to the STROBE guidelines for cross-sectional studies [21] and was designed using a self-administered online questionnaire. The study was conducted at the Nuestra Señora de Candelaria University School of Nursing (EUENSC), which is affiliated with the University of La Laguna (ULL) and located in Tenerife (Canary Islands).

### 2.3. Population and Sample Size

The undergraduate nursing degree curriculum includes clinical placements from the first academic year onwards, delivered across various healthcare settings. The target population comprised all students enrolled in the degree programme (N = 226). Specifically, there were 60 students enrolled in the first year, 50 in the second, 60 in the third, and 56 in the fourth year, corresponding to the 2024–2025 academic year.

The sample size was calculated using the formula for finite populations, assuming a 95% confidence level and a 5% margin of error. A stratified sampling approach was employed, with stratification by academic year to ensure proportional representation across the four years of the Nursing Degree programme, as shown in Table 1. All students within each stratum were given an equal opportunity to participate, as the questionnaire was made available to the entire student body during the data collection phase.

### 2.4. Study Inclusion and Exclusion Criteria

The inclusion criteria for participation in this study were: being a student enrolled in the Nursing Degree programme at EUENSC during the 2024–2025 academic year and voluntary participation in the study. Exclusion criteria included declining to participate and submitting the questionnaire using an email address other than their official institutional account provided by the University of La Laguna (ULL).

### 2.5. Data Collection Instrument

An ad hoc, online, self-administered questionnaire was developed as the measurement instrument, without prior piloting. The questionnaire was designed following a review of the scientific literature, specifically tailored to address the study objectives, and subsequently reviewed and validated by a panel of subject-matter experts. The conceptual and methodological framework for the questionnaire was based on the validated instrument proposed by the joint programme of the ILO, WHO, ICN, and Public Services International (PSI) for violence in the healthcare sector [22]. This panel consisted of three senior academic staff members, all of whom hold doctoral degrees and postgraduate qualifications in occupational health. Content validity was established through expert consensus, and the final version of the questionnaire was agreed upon by the research team.

The questionnaire consisted of 20 closed-ended questions, of which 19 were single-response items and one allowed for multiple responses (type(s) of aggression experienced or witnessed). Several items were measured using a 3- or 4-point Likert scale, intentionally designed without a neutral option. Three questions captured the independent or explanatory variables of a sociodemographic nature: age (ordinal categorical, presented in ranges to prevent identification and preserve participant anonymity), gender (nominal categorical), and academic year (ordinal categorical).

The response options were designed to be simple and quick to complete, thereby increasing the response rate. The questionnaire covered the three main dimensions under study:

#### 2.5.1. Dimension: Knowledge

General knowledge of the phenomenon was explored through questions addressing the concept and types of aggression, existing protocols, and general preventive measures, as well as knowledge of reporting procedures and other institutional resources. The following items collected this information:

Q1: Have you received any training on the prevention of aggression in healthcare settings?

Q5: At the start of your clinical placement, did you receive safety guidelines from academic staff or unit supervisors on how to prevent possible aggression?

Q6: Are you aware of any protocols on the prevention of aggression towards healthcare professionals issued by the Canary Islands Health Service?

Q7: Do you know the definition of aggression in a healthcare setting?

Q8: Are you familiar with the different types of aggression that may occur in healthcare environments?

Q12: Do you know the basic elements required to file a formal complaint in the event of an assault?

As the nursing curriculum does not include formal training on workplace aggression, all items were designed to explore different aspects of the phenomenon as it affects healthcare professionals and/or students within clinical settings. These six items were assessed using a 3-point Likert scale with the following response options: yes, no, and don’t know/no response.

#### 2.5.2. Dimension: Perception

Students’ perceptions of the phenomenon were assessed through questions addressing its perceived importance, the role of students, the prevailing safety culture, and the need for training, among other aspects. The items representing this dimension were:

Q2: Do you think training is necessary for managing aggressive situations?

Q3: If you answered yes to the previous question, in which academic year should it be offered?

Q4: If you were to experience or witness an aggression, how prepared would you feel to manage it?

Q9: How would you rate the safety culture regarding the prevention of aggression in your clinical placement setting?

Q10: Do you believe that aggression towards healthcare professionals is a widespread problem in the sector?

Q11: What role do you think nursing students should play in promoting a safe working environment?

Following consensus by the panel of experts and the research team, item Q10 was selected as the dependent variable for the analysis of associations between variables.

Two of these questions were assessed using a three-point Likert scale: ‘yes’, ‘no’, and ‘don’t know/no response’ (Q2); and ‘passive’, ‘active’, and ‘very active’ (Q11). The remaining questions used a four-point Likert scale as follows:–Q3: Year 1, Year 2, Year 3, Year 4–Q4: Very poorly prepared, Poorly prepared, Well prepared, Very well prepared–Q9: Poor, Average, Good, Very good–Q10: Strongly disagree, Disagree, Agree, Strongly agree

#### 2.5.3. Dimension: Experience

Experiences of aggression during clinical placements were explored through questions addressing either directly experienced incidents or witnessed incidents. The items associated with this dimension were:

Q13: During your clinical placements, have you witnessed or experienced any aggression towards a healthcare professional?

Q14: If you answered yes to the previous question, what type of aggression did you witness or experience?

Q15: If you witnessed an incident of aggression as a nursing student, in which setting did it occur?

Q16: Who was the aggressor?

Q17: Did the professional or yourself have access to any nearby security device?

Item Q13 was selected by consensus as the dependent variable for statistical association analysis.

In this section, only Q15 had a dichotomous response: primary care or hospital care. The remaining items were answered using a four-point Likert scale and single response, with the exception of Q14, which allowed for multiple responses. The response options were as follows:–Q13: Yes, I witnessed it towards another professional/Yes, I experienced it myself/No/Don’t know/No response–Q14: Don’t know or prefer not to say/Physical aggression/Sexual aggression/Verbal aggression–Q16: Patient/Patient’s relative or companion/Healthcare professional/Other–Q17: Yes/No/Don’t know/No response/Unaware of such a device

### 2.6. Data Collection

The questionnaire was distributed via institutional email through the ULL virtual campus, using an online form format, to all students enrolled in the Nursing Degree programme at EUENSC. The questionnaire was available in the first weeks of April, coinciding with the traineeship period of most of the students.

A member of the research team met with each academic year group in their respective classrooms to invite voluntary participation. Students were given instructions on how to complete the questionnaire, including guidance on how to access and verify the form via the link sent to their institutional email account.

### 2.7. Ethics and Confidentiality

This study was conducted in accordance with the principles outlined in the Declaration of Helsinki and with all applicable legal and regulatory requirements in force within Europe and Spain. The processing, communication, and handling of personal data from all participants complied with the provisions of Organic Law 3/2018 of 5 December on the Protection of Personal Data and the Guarantee of Digital Rights [23], as well as Regulation (EU) 2016/679 of the European Parliament and of the Council of 27 April 2016 (General Data Protection Regulation–GDPR) [24], regarding the protection of natural persons with regard to the processing of personal data and the free movement of such data.

The questionnaire included a checkbox for participants to confirm their informed consent to take part in the study.

To ensure the confidentiality of participants’ data, all information was dissociated and anonymised. Only two members of the research team and the Ethics Committee for Research with medicinal products (ECRmp) had access to the original data. Coding was carried out based on the order of questionnaire completion and submission, and each participant was assigned a unique random code generated using Microsoft Excel^®^ spreadsheet software.

The study was approved by the ECRmp under the registration code CHUNSC_2025_23.

### 2.8. Data Processing and Analysis

The data collected and coded in Excel^®^ from the 136 participants were analysed using the advanced statistical software IBM SPSS Statistics^®^ for Windows, version 29.0 (IBM Corp., Armonk, NY, USA).

Descriptive statistics were calculated, including frequencies and percentages for the sociodemographic variables. Contingency tables were generated to present categorical data as frequency counts and to explore potential associations.

A chi-squared statistical analysis was also performed to examine the associations between the independent variables—age, gender, and academic year—and the two dimensions under study: perception and experience of aggression.

## 3. Results

### 3.1. General Knowledge Among Students

Regarding specific training on the prevention of aggression in healthcare settings, only 14% of nursing students reported having received such training, whereas 82.4% stated that they had not accessed this type of instruction prior to beginning their clinical placements.

In addition, only 9.6% of students indicated that they had received safety guidance from teaching staff or from supervisors or deputy managers at the various healthcare centres, compared to 67.6% who had not received any orientation in this regard.

With respect to knowledge of the current prevention protocols issued by the Canary Islands Health Service, only 14.7% of participants reported being familiar with them, while 77.2% stated they were unaware of their existence. Furthermore, 51% reported not knowing about the available security measures in healthcare settings—such as the panic button—and 9.4% selected ‘don’t know/no response’ when asked whether the assaulted individual had access to any on-site security support.

Additionally, 61.18% of students indicated that they were familiar with the definition of aggression in a healthcare context, whereas 27.9% admitted to not knowing the definition. A total of 37.5% reported being aware of the different types of aggression that may occur in healthcare environments, while 49.3% acknowledged that they were unable to distinguish them accurately.

Finally, only 6.6% of students reported knowing the basic requirements for filing a formal complaint in the event of an assault, while 87.5% stated that they were unaware of the relevant procedures.

To facilitate understanding, the main results concerning students’ general knowledge of workplace aggression are summarised in Table 2.

### 3.2. Students’ Perceptions

According to the data gathered on students’ perceptions of aggression in healthcare settings, as shown in Table 3, the majority of nursing students considered aggression towards healthcare professionals to be a widespread problem. Specifically, 40.4% stated they ‘strongly agreed’ and 53.7% ‘agreed’ with this statement, while only 3.7% and 1.5% reported ‘disagreed’ and ‘strongly disagreed’, respectively.

Regarding the need for specific training, 99.3% of respondents expressed that it is necessary to receive training on how to manage aggressive situations. Among those who responded affirmatively, 43% believed this training should be delivered during the first academic year, 23% during the second, another 23% during the third, and 11% in the fourth year.

With respect to self-assessed preparedness to deal with aggression in healthcare settings, 11% rated their level as well prepared, 2.2% as very well prepared, 68.4% as poorly prepared, and 18.4% as very poorly prepared.

In terms of students’ perceptions of the safety culture in their clinical placement settings, 58.8% rated it as ‘average’, 25% as ‘good’, and 8.8% and 2.2% evaluated it as ‘poor’ and ‘very poor’, respectively.

When asked about the role nursing students should play in promoting a safe working environment, 53.7% believed their involvement should be ‘active’ and 44.1% very active’, while only 1.5% considered it to be a ‘passive’ responsibility.

With regard to students’ gender and based on question Q10 (Do you believe that aggression towards healthcare professionals is a widespread problem in the sector?), the majority of female respondents agreed (41.9%) or strongly agreed (36%), with only a small proportion expressing disagreement (4.4%). Male students also perceived aggression as a problem, with 11% agreeing and 4.4% strongly agreeing, while 0.7% registered disagreement. A single respondent identifying as non-binary selected ‘agree’ (0.7%)

No statistically significant associations were observed between the perception of aggression as a widespread issue (using Q10 as the outcome variable) and any of the sociodemographic variables analysed.

### 3.3. Students’ Experiences of Aggression

Among the study participants, 22.8% reported having witnessed an aggression during their clinical placements, 8.1% experienced aggression directly, 41.2% stated they had neither witnessed nor experienced any such event, and 27.9% responded ‘don’t know/no response’.

With regard to the types of aggression witnessed or experienced, 11.5% were physical, 1.6% sexual, 55.8% verbal, and 1.6% responded ‘don’t know/no response’. The setting in which students most frequently witnessed or experienced aggression was the hospital environment (66.7%), compared to 33.3% in primary care settings.

As for the perpetrators, patients were identified as the main aggressors in 73.2% of cases, followed by patients’ relatives or companions (14.6%), other healthcare professionals (9.8%), and others (2.4%), who could not be identified.

In response to the question about whether any security devices or personnel were available during the incident, 51% of participants stated that they were unaware of existing security resources, 9.4% responded ‘don’t know/no response’, 26.4% answered ‘no’, and only 13.2% reported having access to some form of security support (see Table 4).

The data on experiences of aggression by gender, based on question Q13 (During your clinical placements, have you witnessed or experienced any aggression towards a healthcare professional?), revealed the following: among female respondents (n = 113), 38.9% reported neither having experienced nor witnessed any form of aggression, 24.8% stated they had witnessed aggression directed at another professional, and 5.9% had personally experienced it. A further 24.3% of women responded ‘don’t know’ or did not answer.

When analysing the relationship between experiences of aggression (Q13) by gender, 41.2% of students reported no experience of aggression, 22.8% had witnessed an incident, and 8.1% had experienced aggression directly. Of those who had suffered aggression firsthand, 72.7% (n = 8) identified as female.

No statistically significant association was found between experiences of aggression and gender, nor between experiences of aggression and age group. However, a statistically significant association was identified between experiences of aggression and academic year (see Table 5).

## 4. Discussion

The findings of this study reveal a high level of awareness among nursing students regarding the issue of aggression towards healthcare professionals, despite the absence of formal training. Nevertheless, a significant gap was observed in students’ perceived preparedness to handle such situations, with only a small proportion reporting that they felt well prepared—an outcome consistent with numerous previous studies [14,15,16,17,18,19].

A particularly noteworthy finding is the proactive attitude expressed by most students, who indicated their willingness to play an active role in promoting safe environments. This represents a valuable opportunity, which should be supported through targeted training strategies and effective protection policies, both within academic institutions and healthcare settings, as proposed in several studies [14,15,16,17,18,19].

Students’ familiarity with basic concepts related to aggression—such as the definition of aggression in healthcare settings and the different types of violence—was moderately adequate. However, knowledge of the legal procedures required to file a formal complaint was notably low (6.6%), which may partly explain the low reporting rate in contrast to the number of incidents observed, as noted in various studies [15,17,18,25]. This deficiency highlights the urgent need to incorporate specific content on managing violent situations into the Nursing Degree curriculum and other institutional programmes.

Regarding experiences of aggression, the data indicate that risk increases as students’ progress through their academic training, with a statistically significant association found, likely due to the differences in clinical placement hours across the second, third and fourth years.

However, this interpretation must be approached with caution due to the underrepresentation of first- and third-year students in the sample, which may have introduced selection bias and affected the generalisability of the results. The lack of a balanced sample raises the risk that the results disproportionately reflect the characteristics of the more heavily represented academic years, thereby reducing the external validity and generalisability of the study outcomes.

The study found that over 30% of students reported having witnessed or experienced aggression, and 27.9% responded ‘don’t know/no response’. This suggests that the actual rate may be even higher, particularly considering that the study was conducted prior to the first-year students undertaking their initial clinical placements. This is a highly significant and concerning finding, as it confirms—as reported in this and other studies [14,15,16,17,19,20,22,23,24,25,26]—that violence affects not only practising professionals, but also students in training, who should be able to consider clinical environments as safe spaces for learning.

When combined with students’ negative perception of the safety culture in clinical placements and their lack of awareness of available security resources, these findings may reflect the absence of visible protocols, limited training received, or an organisational culture that downplays the issue. These observations are consistent with other studies that emphasise the importance of institutional prevention strategies to improve the response to violence in the healthcare sector [16,17,19,25].

According to multiple studies and recent data from the Spanish Ministry of Health [5,13,15], the main aggressor is typically the patient (73.2%). However, one study identified patients’ relatives and/or companions as the most frequent perpetrators, with verbal aggression being the most common type (55.8%) [18].

Although less frequently reported, aggression by fellow healthcare professionals should not be overlooked. This type of aggression was evident in the present findings and has also been documented in other studies [15,17,18,25,26].

Among the available scientific literature, only one article [18] aligns with our finding that the hospital setting is the most frequent context in which aggression occurs. This may be due to the significantly higher number of placement hours assigned to hospital-based training compared to primary care placements in the nursing curriculum.

Furthermore, it is worth noting that first-year students are typically introduced to primary care placements for the first time, and, at the time of this study, had not yet undertaken the relevant clinical training module, as previously mentioned.

Taken together, these findings underscore the need to strengthen theoretical and practical training in aggression prevention, enhance the visibility and dissemination of institutional protocols, and foster a stronger safety culture within healthcare environments.

In terms of gender, 83.09% of the student sample were women, which may have influenced the conclusion that gender plays a role in the perception of aggression.

This study has several limitations. Although the response rate was relatively high (60.18%), there is a potential for selection bias, as the views and experiences of non-respondents may differ from those who participated. In addition, the calculated sample size was not fully achieved, limiting the generalisability of the findings. The specific sociocultural context must also be considered. A key limitation of this study was the absence of a validated instrument. Combined with the small sample size and the descriptive scope of the research, this prevented a full psychometric analysis, which hinders comparisons with other studies. Although content validation was carried out by subject-matter experts, the lack of evidence regarding structural validity and internal consistency may affect the interpretation of the results. We recommend that future studies validate the instrument using larger and more diverse samples. Lastly, surveys relying on participants’ recall of past experiences may be subject to memory bias, particularly when addressing emotionally significant events such as aggression.

### Future Lines of Research

Based on the results obtained, it would be appropriate to develop longitudinal studies to examine nursing students’ level of knowledge, risk perception, and experiences of aggression throughout their academic training. Additionally, multicentre studies are recommended, involving various institutions and healthcare and sociocultural contexts, with the aim of comparing safety culture across different clinical placement settings.

Another promising line of inquiry would be to assess the effectiveness of specific educational interventions for the prevention of aggression. This could be achieved through controlled trials or structured training programmes, evaluating the impact on participants’ knowledge following the intervention.

Finally, it is essential to further investigate the factors associated with the normalisation of violence in clinical settings. This includes exploring both individual and institutional barriers that hinder reporting and prevention efforts.

## 5. Conclusions

This study reveals significant gaps in the preparedness of nursing students at EUENSC to deal with situations of aggression in healthcare settings. Although the majority perceive violence as a widespread issue, levels of specific training and awareness of institutional response protocols are notably low.

It is particularly concerning that over 30% of participants reported having experienced or witnessed aggression during their clinical placements, within a context marked by a perceived lack of safety culture. This scenario suggests a potential normalisation of violence and highlights a shortfall in effective strategies for prevention and management during the formative stages of professional development.

Preventing violence during clinical training must become an institutional priority if the aim is to build a genuine and lasting culture of safety within the healthcare sector.

In light of the findings, it is essential to advance the development of targeted educational modules on the prevention and management of aggression in healthcare settings, specifically aimed at nursing students. It is therefore proposed that this topic be structurally and comprehensively integrated into undergraduate nursing curricula, enabling future healthcare professionals to acquire not only clinical skills but also legal knowledge and self-care strategies in situations involving violence. This integration should go beyond mere awareness-raising and include modules designed with active pedagogical approaches, such as simulation-based learning, problem-based learning, role play, guided reflection, and clinical debriefing.

Equally important is recognising the strategic role of the clinical mentor or supervisor, who serves as a professional role model and guide in the students’ transition into clinical practice. Mentors not only support clinical learning but also play a key role in the early identification of violent incidents, in offering emotional support to affected students, and in ensuring appropriate reporting and response mechanisms.

The coordinated implementation of these educational resources, together with the strengthening of clinical mentors’ roles, will contribute to fostering an institutional culture of zero tolerance towards violence in healthcare settings.

## Figures and Tables

**Table 1 nursrep-15-00245-t001:** Sample size calculation and final sample obtained by academic year.

Class Year	*n*	*n* Estimated ^1^	*n* Observed ^2^
Year 1	60	38	33
Year 2	50	32	43
Year 3	60	38	19
Year 4	56	35	41

^1^ Estimated refers to the sample size calculated using the finite population formula, adjusted for expected attrition and stratified by academic year. ^2^ Observed indicates the actual sample obtained for this study per academic year.

**Table 2 nursrep-15-00245-t002:** Students’ knowledge of aggression in healthcare settings.

Question Number	n	%
**1. Have you received any training on the prevention of aggression in healthcare settings?**		
Yes	19	14.0
No	112	82.4
Don’t know/No response	5	3.7
**5. At the start of your clinical placement, did you receive safety guidelines from academic staff or unit supervisors on how to prevent possible aggression?**		
Yes	13	9.6
No	92	67.6
Don’t know/No response	29	21.3
**6. Are you aware of any protocols on the prevention of aggression towards healthcare professionals issued by the Canary Islands Health Service?**		
Yes	20	14.7
No	105	77.2
Don’t know/No response	11	8.1
**7. Do you know the definition of aggression in a healthcare setting?**		
Yes	84	61.8
No	38	27.9
Don’t know/No response	14	10.3
**8. Are you familiar with the different types of aggression that may occur in healthcare environments?**		
Yes	51	37.5
No	67	49.3
Don’t know/No response	18	13.2
**12. Do you know the basic elements required to file a formal complaint in the event of an assault as a healthcare professional?**		
Yes	9	6.6
No	119	5.9
Don’t know/No response	8	87.5

**Table 3 nursrep-15-00245-t003:** Students’ perceptions of aggression.

Question Number	n	%
**2. Do you think training is necessary for managing aggressive situations?**		
Yes	135	99.3
No	0	0
Don’t know/No response	1	0.7
**3. If you answered yes to the previous question, in which academic year should it be offered?**		
Year 1	58	43.0
Year 2	31	23.0
Year 3	31	23.0
Year 4	15	11.0
**4. If you were to experience or witness an aggression, how prepared would you feel to manage it?**		
Very poorly prepared	25	18.4
Poorly prepared	93	68.4
Well prepared	15	11.0
Very well prepared	3	2.2
**9. How would you rate the safety culture regarding the prevention of aggression in your clinical placement setting?**		
Poor	12	8.8
Average	80	58.8
Good	34	25.0
Very good	3	2.2
**10. Do you believe that aggression towards healthcare professionals is a widespread problem in the sector?**		
Strongly disagree	2	1.5
Disagree	5	3.7
Agree	73	53.7
Strongly agree	55	40.4
**11. What role do you think nursing students should play in promoting a safe working environment?**		
Passive	2	1.5
Active	73	53.7
Very active	60	44.1

**Table 4 nursrep-15-00245-t004:** Students’ experiences of aggression.

Question Number	n	%
**13. During your clinical placements, have you witnessed or experienced any aggression towards a healthcare professional?**		
Yes, I witnessed it towards another professional	31	22.8
Yes, I experienced it myself	11	8.1
No	56	41.2
Don’t Know/No response	38	27.9
**14. If you answered yes to the previous question, what type of aggression did you witness or experience?**		
Physical aggression	7	11.5
Sexual aggression	1	1.6
Verbal aggression	34	55.8
Don’t know/No response	18	29.5
Other	1	1.6
**15. If you witnessed an incident of aggression as a nursing student, in which setting did it occur?**		
Hospital care	26	66.7
Primary care	13	33.3
**16. Who was the aggressor?**		
Patient’s relative or companion	6	14.6
Patient	30	73.2
Healthcare professional	4	9.8
Other	1	2.4
**17. Did the professional or yourself have access to any nearby security device?**		
Yes	7	13.2
No	14	26.4
Don’t know/No response	5	9.4
Unaware of such a device	27	51.0

**Table 5 nursrep-15-00245-t005:** Association between experiences of aggression and academic year.

	Value	df	Asymptotic Significance (Two-Tailed)
Pearson chi-squared	122.139 ^1^	9	<0.001
Likelihood ratio	126.178	9	<0.001
N of valid cases	136		

^1^ 5 cells (31.3%) have expected count less than 5. The minimum expected count is 1.54.

## Data Availability

The data presented in this study, as well as the pre-registration details, are available from the corresponding author upon reasonable request.

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
