# Peer review of "Nursing Students’ Perceptions and Experiences of Aggression During Clinical Placements"

_nursrep, 2025, doi:10.3390/nursrep15070245_

Round 1
Reviewer 1 Report
Comments and Suggestions for Authors
This study addresses a relevant and timely issue concerning nursing students’ perceptions and experiences of aggression during clinical placements. The manuscript is well-structured and generally clear. However, the following aspects require major revision before it can be considered for publication:
The study uses an ad hoc questionnaire, but no evidence of psychometric validation (e.g., factor analysis, reliability tests) is presented. The absence of structural validity and reliability coefficients weakens the scientific robustness of the findings.
The analysis is limited to descriptive statistics and chi-square tests. The authors are encouraged to consider multivariate analysis to control for potential confounders and better interpret associations. It should also be noted that in one of the main analyses (Table 5), more than 30% of the expected cell counts were below 5, which may compromise the validity of the chi-square test and lead to inflated significance. The authors should address this limitation and consider alternative approaches.
The sample size falls below the calculated estimate, especially among third-year students. The potential for selection bias should be more clearly acknowledged and discussed in the limitations.
The discussion would benefit from deeper comparisons with international literature and studies from similar educational contexts (e.g., Ata & Yılmaz, 2022; Hallett et al., 2021).
Although the overall quality of English is understandable, there are sentences that should be revised for clarity and academic tone.
These improvements would enhance the clarity, validity, and overall impact of the study.
Comments on the Quality of English Language
The manuscript is generally understandable; however, several sentences would benefit from revision to improve clarity and flow. Particular attention should be paid to grammar, word choice, and sentence structure to ensure consistency with academic English standards.
Author Response
This study addresses a relevant and timely issue concerning nursing students’ perceptions and experiences of aggression during clinical placements. The manuscript is well-structured and generally clear. However, the following aspects require major revision before it can be considered for publication:
The study uses an ad hoc questionnaire, but no evidence of psychometric validation (e.g., factor analysis, reliability tests) is presented. The absence of structural validity and reliability coefficients weakens the scientific robustness of the findings.
The analysis is limited to descriptive statistics and chi-square tests. The authors are encouraged to consider multivariate analysis to control potential confounders and better interpret associations. It should also be noted that in one of the main analyses (Table 5), more than 30% of the expected cell counts were below 5, which may compromise the validity of the chi-square test and lead to inflated significance. The authors should address this limitation and consider alternative approaches.
We appreciate the reviewer’s observation regarding the psychometric assessment. We acknowledge that the absence of such analyses limits the robustness of our findings. However, the questionnaire used was developed ad hoc due to the lack of previously validated instruments appropriate to our specific context and population.
To construct the instrument, a literature review was conducted, and content validity was ensured through expert judgement. Given the exploratory and descriptive nature of the study, and the small sample size, psychometric evaluation was not performed. This limitation has been explicitly acknowledged in the revised manuscript (line 402).
We are currently working on the development and validation of an instrument for nursing students as part of a doctoral thesis. The results of this validation process, including the corresponding analyses, will be published in a subsequent study.
The sample size falls below the calculated estimate, especially among third-year students. The potential for selection bias should be more clearly acknowledged and discussed in the limitations.
In addition to including the small sample size bias discussed above, we have clarified this situation, and it has been discussed in lines 361-363. Thank you for your observation. As noted above, the limitations related to the small sample size have been acknowledged, and we have also clarified the underrepresentation of third-year students. This potential selection bias has been explicitly discussed in the revised manuscript (lines 361-363).
The discussion would benefit from deeper comparisons with international literature and studies from similar educational contexts (e.g., Ata & Yılmaz, 2022; Hallett et al., 2021).
Thank you for your suggestion. We have incorporated the studies by Ata & Yılmaz (2022) and Hallett et al. (2021), as well as a more recent publication by Hallett et al. (2023), to strengthen the discussion and contextualise our findings within the broader international literature. These studies are now cited in references 14, 15, and 17, respectively.
Although the overall quality of English is understandable, there are sentences that should be revised for clarity and academic tone.
Thank you for your comment. We would be grateful if you could indicate which specific sentences require revision so that we may address them appropriately.
The manuscript has been reviewed for grammar, punctuation, spelling, and academic style by Dr Alejandro García-Aragón, a professional translator specialising in scientific texts, and by Reece Anderson, a native English-speaking medical translator and editor. A language editing certificate has been attached (code: CrtNrStP&EAgg2).
We remain fully committed to improving the clarity and quality of our manuscript and welcome any additional suggestions you may have.

Reviewer 2 Report
Comments and Suggestions for Authors
Dear Authors,
I would like to offer the following suggestions to further strengthen the clarity, methodological rigor, and practical value of your work:
Abstract: In line 17, I suggest modifying the sentence to: "An ad hoc self-administered questionnaire was distributed among 266 students from all academic years..."
Introduction: Consider including additional empirical data on the prevalence of aggression in clinical settings across Europe or internationally, to strengthen the contextual background and underscore the global relevance of the issue.
Materials and Methods: You mention that the questionnaire was "designed following a review of the scientific literature", but the references to the sources (authors, publication years) are missing. It would be methodologically appropriate to cite the key sources that informed the development of the questionnaire. Additionally, please clarify whether the instrument was pilot tested prior to full-scale distribution.
Results: Consider adding a gender-based breakdown of the data, as gender may play a relevant role in the perception and experience of aggression.
Discussion: The discussion of recommended interventions could benefit from more specificity. For example, you might elaborate on what kind of educational strategies or training programs could be implemented to prepare students for managing aggression.
Limitations: Please briefly address why a standardized instrument for measuring aggression was not used in this study. A short justification would add transparency and strengthen the methodological section.
Conclusions: You may consider concluding with a clear call to action regarding the development of targeted educational modules and the integration of this topic into nursing curricula.
In addition, the conclusion could specify which pedagogical approaches (e.g., simulation training, reflective supervision, debriefing) may be effective, and highlight the important role of mentors and clinical settings in preventing and addressing aggression.
Author Response
Dear Authors,
I would like to offer the following suggestions to further strengthen the clarity, methodological rigor, and practical value of your work:
Abstract: In line 17, I suggest modifying the sentence to: "An ad hoc self-administered questionnaire was distributed among 266 students from all academic years..."
Thank you for your suggestion. The sentence has been revised accordingly to improve clarity and precision in the abstract.
Introduction: Consider including additional empirical data on the prevalence of aggression in clinical settings across Europe or internationally, to strengthen the contextual background and underscore the global relevance of the issue.
Thank you for your valuable suggestion. We have incorporated the systematic review and meta-analysis by Li et al. (2020) (reference 4), which provides international data on the prevalence of aggression in clinical environments. The relevant comparisons have been added in lines 36–43 to enhance the contextualisation and global relevance of the issue.
Materials and Methods: You mention that the questionnaire was "designed following a review of the scientific literature", but the references to the sources (authors, publication years) are missing. It would be methodologically appropriate to cite the key sources that informed the development of the questionnaire. Additionally, please clarify whether the instrument was pilot tested prior to full-scale distribution.
We are grateful for your input. We have now incorporated the sources that informed the development of the questionnaire, including documents from the International Labour Office (ILO), World Health Organization (WHO), International Council of Nurses (ICN), and Public Services International (PSI) (reference 21), as indicated on lines 113–116. Additionally, we have clarified that no pilot testing was conducted prior to the full-scale distribution of the questionnaire (line 111).
Results: Consider adding a gender-based breakdown of the data, as gender may play a relevant role in the perception and experience of aggression.
Thank you for your suggestion. We have included a gender-based breakdown of the data. The influence of gender on the perception of aggression has been discussed in lines 285–290, and its role in the experience of aggression is addressed in lines 315–321. Furthermore, we have highlighted the relevance of this variable in the discussion section on line 396.
Discussion: The discussion of recommended interventions could benefit from more specificity. For example, you might elaborate on what kind of educational strategies or training programs could be implemented to prepare students for managing aggression.
Thank you for your valuable observation. We agree that this information is relevant; however, we believe the coherence of the text improves if it is incorporated into the conclusions section, as suggested below. The strategies and recommendations have been added in lines 436–452.
Limitations: Please briefly address why a standardized instrument for measuring aggression was not used in this study. A short justification would add transparency and strengthen the methodological section.
Thank you for your suggestion. This limitation has been acknowledged and addressed in lines 402–408 to enhance transparency and reinforce the methodological rigour of the study.
Conclusions: You may consider concluding with a clear call to action regarding the development of targeted educational modules and the integration of this topic into nursing curricula. In addition, the conclusion could specify which pedagogical approaches (e.g., simulation training, reflective supervision, debriefing) may be effective, and highlight the important role of mentors and clinical settings in preventing and addressing aggression.
Thank you very much for your valuable suggestion. This reflection, along with the proposed pedagogical approaches, has been incorporated into the conclusion section (lines 436–452). We appreciate the opportunity to improve the manuscript and trust that these additions enhance its clarity, impact, and practical value.
Reviewer 3 Report
Comments and Suggestions for Authors
Intro: Good description of problem of violence towards nurses globally and in Spain
M+M: Aims of the study are clearly stated – might be better in Introduction section. Participant inclusion, questions, data collection, data processing and analysis were all thoroughly described. Note: it appears there were 135 participants, not 136.
Results: Most results were clearly presented in Tables and accompanying text. It would be helpful to give the actual results of in what academic year the experiences of aggression occurred and whether these were largely classroom or clinical years.
Discussion: Authors do a good job reviewing their findings and discussing consistencies with other studies in the literature. Limitations of the study are well described. Noting that nursing students are largely aware of the presence of aggression while there is a lack of training in prevention and reporting, they recommend this training while still in school. Very importantly, the authors emphasize the perceived lack of a safety culture and potential normalization of violence. Often the onus is on the victim to protect her/him-self; the onus should be on the institution and society to provide safe learning and work spaces.
Comments: Training in aggression awareness and prevention is analogous to what was done for blood borne pathogen exposures; instilling this training while in formative years can be very effective. The authors are correct in emphasizing that their “findings underscore the need to strengthen theoretical and practical training in aggression prevention, enhance the visibility and dissemination of institutional protocols, and foster a stronger safety culture within healthcare environments.”
Author Response
Intro: Good description of problem of violence towards nurses globally and in Spain.
M+M: Aims of the study are clearly stated – might be better in Introduction section. Participant inclusion, questions, data collection, data processing and analysis were all thoroughly described. Note: it appears there were 135 participants, not 136.
Thank you for your careful review. The total number of participants was indeed 136, as shown in Table 5. Upon review, we noted that one response was missing from Table 1 (year 4), which may have caused the confusion. Not all participants answered every question, but the total number is reflected in those items where complete data were available, as seen in questions 1, 6, 7, 8, and 12 (Table 2); questions 2 and 4 (Table 3); and question 13 (Table 4).
Results: Most results were clearly presented in Tables and accompanying text. It would be helpful to give the actual results of in what academic year the experiences of aggression occurred and whether these were largely classroom or clinical years.
Thank you for your observation. The academic year in which the aggressions occurred was not specifically studied. Instead, participants were asked to indicate the year of their degree during which the incident took place while on clinical placement.
Discussion: Authors do a good job reviewing their findings and discussing consistencies with other studies in the literature. Limitations of the study are well described. Noting that nursing students are largely aware of the presence of aggression while there is a lack of training in prevention and reporting, they recommend this training while still in school. Very importantly, the authors emphasize the perceived lack of a safety culture and potential normalization of violence. Often the onus is on the victim to protect her/him-self; the onus should be on the institution and society to provide safe learning and work spaces.
We very much appreciate your thoughtful assessment and support of our discussion. Thank you for highlighting the importance of institutional and societal responsibility in ensuring safe learning and working environments.
Comments: Training in aggression awareness and prevention is analogous to what was done for blood borne pathogen exposures; instilling this training while in formative years can be very effective. The authors are correct in emphasizing that their “findings underscore the need to strengthen theoretical and practical training in aggression prevention, enhance the visibility and dissemination of institutional protocols, and foster a stronger safety culture within healthcare environments.”
We welcome and deeply value your thoughtful reflection. We have incorporated the importance of integrating such training into nursing curricula, as well as the use of methodological approaches that reinforce learning in this area, as stated in lines 436–452. We appreciate your support and hope that our contributions enhance the quality, transparency, and impact of the study.